# Sarcopenia Worsening One Month after Transarterial Radioembolization Predicts Progressive Disease in Patients with Advanced Hepatocellular Carcinoma

**DOI:** 10.3390/biology10080728

**Published:** 2021-07-30

**Authors:** Giulio Eugenio Vallati, Claudio Trobiani, Leonardo Teodoli, Quirino Lai, Federico Cappelli, Sara Ungania, Carlo Catalano, Pierleone Lucatelli

**Affiliations:** 1Interventional Radiology Unit of “IRCCS Istituto Nazionale Tumori Regina Elena”, 00138 Rome, Italy; giulio.vallati@ifo.gov.it (G.E.V.); federicocappelli@hotmail.it (F.C.); 2Department of Radiological Sciences, Oncology and Anatomical Pathology, Policlinico Umberto I, University of Rome “Sapienza”, 00161 Rome, Italy; l.teodoli@gmail.com (L.T.); carlo.catalano@uniroma1.it (C.C.); pierleone.lucatelli@gmail.com (P.L.); 3Department of General Surgery and Organ Transplantation, Sapienza University of Rome, 00161 Rome, Italy; lai.quirino@libero.it; 4Physics Department of “Istituto Regina Elena, Istituto di Ricovero e Cura a Carattere Scientifico”, 00138 Rome, Italy; sara.ungania@gmail.com

**Keywords:** TARE, sarcopenia, HCC, interventional oncology, liver

## Abstract

**Simple Summary:**

Sarcopenia measured at one-month CT follow up after TARE (transarterial radioembolization) treatment is a predictive factor for the best tumor response in patients with locally advanced HCC.

**Abstract:**

(1) Background: To demonstrate correlation between skeletal muscle depletion measured before and after one month of TARE treatment and its induced local response rate. (2) Material and methods: For this retrospective, single center study, we evaluated 86 patients with HCC treated with TARE. Sarcopenia status was measured using the psoas muscle index (PMI). The PMI was calculated according to the formula: PMI [mm/m^2^]: [(minor diameter of left psoas + major diameter of left psoas + minor diameter of right psoas + major diameter of right psoas)/4]/height in m^2^. Population was divided in two groups according to the delta value of PMI measured at the time of TARE and one month after TARE, a group in which the delta PMI was stable or increased (No-Sarcopenia group; *n* = 42) vs. a group in which the delta-PMI decreased (Sarcopenia group; *n* = 44). Patient response was evaluated at 1, 3 and 6 months after TARE treatment with CT/MRI. (3) Results: When the radiological response of the tumor was evaluated according to the mRECIST criteria, the two groups were similar in terms of rates of complete response (*p* = 0.42), partial response (*p* = 0.26) and stable disease (*p* = 0.59). Progressive disease (PD) was more commonly observed in the Sarcopenia group (38.6% vs. 11.9%; *p* = 0.006). (4) Conclusions: Worsening of sarcopenia status measured one month after TARE is able to predict patients who will undergo disease progression.

## 1. Introduction

In patients with hepatocellular carcinoma (HCC), curative surgical approaches (i.e., resection and liver transplantation) are often not feasible due to the locally advanced stage of the disease [1,2]. These patients, according to international guidelines [3], can be treated with catheter-based treatments as transarterial chemoembolization (TACE) or transarterial radioembolization (TARE). These treatments have evolved as a safe and effective treatment modality, providing high rates of local tumor control and a low risk of major systemic side effects [4]. TARE consist in the administration, via the standard arterial route, of a radioisotope (Yttrium, Y90), whose dose is personalized according to total tumor burden and hepatic volumetry on the basis of a computed tomography (CT) preprocedural study.

The anti-tumoral action is determined principally by the radioisotopes beta emission that is able to penetrate tumoral tissue by a median of 2.5 mm; moreover, microspheres that vehicle the radioisotope can act a bland embolization. Post-TARE treatment response is usually evaluated through CT, FDG-PET, or MRI. However, the exact timing to assess treatment efficacy (local tumor control) differs from TACE procedures, that are usually reevaluated at 30 days. In the case of TARE treatments, due to the radical changes observed in liver parenchyma as a consequence of the radiation response that can last for months, tumor response is usually assessed at a later stage, usually 3–6 months from the treatment session. In addition, current methods of response evaluation have a lag from three to six months to determine treatment response, thus leading to a delayed therapy modification [5]. As a consequence, a prompt assessment of early treatment response is not currently available.

Sarcopenia has been defined by the Special Interest Group of the European Sarcopenia Working Group in 2010 as the “progressive loss of muscle mass and strength with a risk of adverse outcomes such as disability, poor quality of life and death” [6]. The prognostic role of sarcopenia status in HCC patients, either for those who underwent surgical resection or medical treatment (sorafenib), has been shown in different studies [7,8,9]. Anyhow, only few studies have discussed the prognostic role of sarcopenia in HCC patients undergoing trans-catheter intra-arterial therapies such as TACE and TARE [10] and none of them analyzed its role as a negative predictor of local response after TARE treatment.

The aim of this study is to evaluate a potential correlation between skeletal muscle depletion measured before and after one month from TARE treatment and its induced local response rate, in order to understand if sarcopenia worsening can be used to promptly identify a sub-cohort of patients at high risk for progressive disease that could benefit an early locoregional retreatment, a nutritional support or further medical strategies [11,12].

## 2. Materials and Methods

### 2.1. Patients

We performed a retrospective analysis of 86 patients with HCC treated with TARE during the period 1 January 2012–31 May 2020. All the patients received an intermediate evaluation one month after the TARE procedure and a final assessment of the radiological response three months after TARE. According to the radiological findings, the entire population was divided in two groups according to the delta value of the psoas muscle index (PMI) measured at the time of TARE and one month after TARE. The groups were composed by patients in which the delta PMI was stable or increased (No-Sarcopenia group; *n* = 42) vs. patients in which the delta-PMI decreased (Sarcopenia group; *n* = 44). The median follow-up period after the last radiological assessment after TARE was six months (IQR = 0–9). The study received local ethic committee approval.

### 2.2. TARE Treatment and Its Efficacy Assessment

All procedures were performed by one experienced (>15 years) interventional radiologist via the femoral access. Detailed hepatic anatomy was assessed by performing a digital subtracted angiography (DSA) of the proper hepatic artery; this permitted the identification of arterial branches feeding the area of the liver parenchyma target of the radioembolization treatment.

In all cases, a treatment simulation by injection, via a coaxial 2.7-F micro-catheter (Progreat; Terumo Europe NV, Leuven, Belgium) positioned in the appropriate feeder artery, of technetium-99 m (99 mTc) macro-aggregated albumin (MAA) was performed and evaluated by SPECT imaging (15). With a median interlapse of 8 days from the simulation, radioembolization treatment was performed by positioning the coaxial microcatheter in the same injection position. A post therapy SPECT/CT (Symbia IntevoTM system; Siemens, Erlangen, Germany) scan was performed between 1 and 20 h after SIRT to evaluate the 90Y-microspheres distribution. Post therapy SPECT/CT imaging was used to perform 2D and 3D dosimetry as described below.

As a first step, the accuracy and intensity of 90Y-microspheres activity distribution was evaluated through the 2D activity intensity peak (pixel value) of the signal along a line crossing the treated area. The higher the peak, the more intense is the signal inside the treated area.

Then, 3D effective dose in Gy delivered to the target and normal liver per unit cumulated activity (GBq) was calculated based on the activity distribution on SPECT/CT imaging. Lesion and normal liver were delineated on an MIM 6.1.7 workstation (MIM Software Inc., Cleveland, OH, USA) and dose calculation was performed on such volumes. For each patient, mean absorbed dose (<D>) in Gy obtained for normal liver and tumor were compared with expected ones (<D> to tumor > 100 Gy, <D> to normal liver < 40 Gy).

Assessment of treatment response was categorized according to mRECIST criteria [13] at 3 and 6 months after both typologies of treatment.

### 2.3. Definition of Sarcopenia Measurement 

PMI was determined by measuring the major diameter and the minor diameter (measured perpendicularly to the major) of the right and of the left psoas on an axial CT scan at the time of TARE (Figure 1). A follow-up scan after TARE therapy for the assessment of PMI modification was performed at one, three and six months after the procedure (Figure 2) [14].

Sarcopenia status was defined as a decrease in the delta PMI calculated comparing the PMI values at the time of TARE and one month after TARE.

### 2.4. Statistical Analysis

Continuous variables were reported as medians and interquartile ranges (IQR). Dichotomous variables were reported as numbers and percentages. No missing data were present in the evaluated population. The Mann–Whitney *U* test and Fisher’s exact test were used for comparisons of continuous and categorical variables, respectively.

A Cox regression model was constructed to identify the risk factors for progressive disease after TARE. Hazard ratios (HR) and 95% confidence intervals (95%CI) were reported.

The accuracy of the different sarcopenia measurements performed over time was assessed through c-statistics analysis, with the intent to evaluate their ability to predict the progressive disease after TARE. Areas under the curve (AUCs) and 95% CIs were reported. The PMI at the time of TARE was compared with the PMI measurements at one and three months after TARE and with the delta-PMI measurements between the first PMI measurement and the controls at one and three months. Time-to-progression probabilities were estimated using the Kaplan–Meier method. Time-to-progression rates comparisons were estimated using the log-rank method. The starting time-point for assessing the time-to-progression survival analysis corresponded to the time of radiological assessment performed 3 months after TARE.

Variables with a *p* < 0.05 were considered statistically significant. We used the SPSS statistical package version 24.0 (SPSS Inc., Chicago, IL, USA).

## 3. Results

Patient-, tumor- and treatment-related characteristics are reported in Table 1. No differences were observed between the two groups in terms of demographic aspects of the patients. Median age (*p* = 0.45) and male sex prevalence (*p* = 0.62) were similar. In addition, tumor characteristics evaluated at the time of TARE were not statistically significant. The diameter of the target lesion (*p* = 0.51), the number of nodules (*p* = 0.48), the liver involvement (*p* = 0.46), the macrovascular invasion (*p* = 1.00) and the presence of nodal metastases (*p* = 0.60) were not statistically relevant. In addition, the HCC biological markers PIVKA (*p* = 0.91) and alpha-fetoprotein (*p* = 0.87) were not statistically relevant.

The only significant statistical difference was observed in terms of sarcopenia status. Paradoxically, the Sarcopenia group initially presented a slightly better sarcopenic status, with a median PMI = 11.3 mm/m^2^ vs. 10.2 mm/m^2^ (*p* = 0.04).

One month after TARE, the median PMI became similar between the two groups (10.7 mm/m^2^; *p* = 0.70). This datum explained the median delta-PMI observed comparing the results at the time of TARE and one month after the radiological procedure. In the No-Sarcopenia group, the median delta-value was 0.1 vs. −0.4 observed in the Sarcopenia group (*p* < 0.0001). In other terms, 25 (29.1%) patients showed an increased, 17 (19.8%) a stable and 44 (51.2%) a decreased PMI value. In Figure 3, a more detailed distribution of the entire population according to the initial PMI value and the delta-PMI at one month was reported.

After three months, when the final radiological assessment was performed, the spread between the two groups further increased, with a median delta PMI of 0.1 vs. –0.6 in the No-Sarcopenia vs. the Sarcopenia group (*p* < 0.0001).

Interestingly, the delta PMI presented a similar modification across the different time-points of measurement, no matter the sex of the patient (Table 1), showing that the PMI decline showed similar attitudes in men and women.

At the radiological response evaluation of the tumor according to the mRECIST criteria, the two groups were similar in terms of rates of complete response (*p* = 0.42), partial response (*p* = 0.26) and stable disease (*p* = 0.59). Only the progressive disease (PD) was more commonly observed in the Sarcopenia group (38.6% vs. 11.9%; *p* = 0.006). When both intra- and extra-hepatic PD was evaluated, also in this case, a greater percentage of cases was observed in the Sarcopenia group (45.5% vs. 16.7%; *p* = 0.005). Dose distribution and m-RECIST response were evaluated through SPECT/CT scan and multiphasic CT, as shown in Figure 4.

### 3.1. Risk Factors for Progressive Disease

Following multivariable Cox regression analysis for the risk of PD, classical risk factors for tumor progression failed to be statistically relevant when the delta-PMI at one month was added among the variables. Interestingly, only the delta-PMI at one month was statistically significant as an independent risk factor for PD, with an HR = 3.25 (95%CI = 1.35–7.81; *p* = 0.009). As previously reported, the dimension and the number of lesions, the presence of macrovascular invasion or the presence for nodal metastases were not statistically significant (Table 2).

### 3.2. Diagnostic Ability of the Sarcopenia Measurement

The diagnostic ability for progressive disease of the sarcopenia measurement is reported in Table 3. It was interesting to note that the initial sarcopenia assessment at the time of TARE had no diagnostic ability, with an AUC = 0.59 (*p* = 0.19). This ability progressively increased when the time passed from the TARE procedure. At one month, the PMI value had an AUC = 0.65 (*p* = 0.03). After three months, the PMI value presented an AUC = 0.73 (*p* = 0.001). The delta values were also statistically relevant in the diagnosis of PD, with the delta-PMI at one month presenting an AUC = 0.70 (*p* = 0.003) and the delta-PMI at three months an AUC = 0.81 (*p* < 0.0001).

### 3.3. Time-to-Progress Rates

Observing the time-to-progress rates between the No-Sarcopenia and Sarcopenia groups, it was interesting to observe that the No-Sarcopenia group reported 6- and 12-month rates of 14.5% and 19.5%, while the Sarcopenia group presented 6- and 12-month rates of 43.4% and 46.9%, with a log-rank *p* = 0.005 (Figure 5).

A sub-analysis performed in men and women showed similar results. In men, the No-Sarcopenia group reported 6- and 12-month time-to-progress rates of 15.4% and 21.0%, while the Sarcopenia group presented 6- and 12-month rates of 41.0% and 45.5% (log-rank *p* = 0.003). In women, the No-Sarcopenia group reported 6- and 12-month time-to-progress rates of 11.1% and 11.0%, while the Sarcopenia group presented 6- and 12-month rates of 50.0% and 50.0% (log-rank *p* = 0.007).

## 4. Discussion

These study results report the ability of sequential sarcopenia measurement in patients undergoing TARE treatment for HCC in predicting the occurrence of post-procedural progressive disease. This concept is new in the setting of loco-regional therapies and could lead to several clinical implications in the setting of cirrhotic patients with locally advanced HCC.

Sarcopenia status difference measured between pre-procedural CT and one-month post-TARE CT more commonly worsened in the group of patients experiencing a progressive disease, meanwhile it did not show any significant modification in patients whose responses were categorized as complete, partial or stable. This discrepancy was even more evident at the three-month evaluation, with the skeletal muscle depletion continuing to diverge between the two cohorts. Moreover, this attitude was similar in men and women, no matter the initial morphological discrepancies existent between the different sexes.

This fact further strengthens the ability of skeletal muscle depletion to precociously predict at one month, in a non-invasive method, an event (namely, progressive disease) affecting the patient in the following months. On the other hand, pre-procedural sarcopenia status did not show any statically significant differences between the group of patients who underwent to progressive disease vs. the ones who underwent complete response, partial response and stable disease, meaning that the initial sarcopenia evaluation cannot be used as a useful predictor of poor post-TARE oncological response. Sarcopenia measurement could therefore aid in the identification of cohorts of patients that will suffer progression of disease.

Until now, sarcopenia has been used as a prognostic factor in different clinical settings. As for liver diseases, sarcopenia has been used as prognostic factor in HCC patients [7], HCC patients undergoing sorafenib systemic treatment [8,9], HCC patients undergoing intra-arterial chemoembolization treatment [10] and cirrhotic patients, either in a pre- or post-transplant setting [15]. A recent study also suggests that sarcopenia might be associated with an increased risk of HCC among male patients with cirrhosis [16]. Even though different roles for sarcopenia have been established in the setting of cirrhotic and HCC patients, there are no current studies that have proven a correlation between an early (one-month) skeletal muscle depletion and post-TARE local response rate.

Response evaluation to TARE treatment is conducted using CT, MRI, or FDG-PET. All these methods have a lag of at least three months for giving correct information about TARE treatment outcomes [5]. Recently, a post-procedural SPECT/CT distribution analysis has been proposed as a method for early evaluation of post-TARE response, but this new method has not still gained a definitive role [15,16]. Up to now, there is no early (one-month) evaluation method of TARE treatment response. In particular, there is no easily evaluable method to promptly evaluate patients who will suffer progression of HCC disease. Sarcopenia status difference, measured one month after procedure on routinely performed multiphasic CT scan, can be easily used as a non-invasive method of identification of patients with high-risk for progressive disease.

Moreover, the ability to early predict patients at risk for progressive disease is a crucial information, potentially impacting the clinical management of these patients.

First, patients who show an early modification of skeletal muscle depletion benefit from nutritional support and physical therapy [11,17], that are known as a cornerstone in the management of sarcopenia. Recent research in the field aimed to overcome skeletal muscle depletion using multimodal interventions, including exercise, nutrition and ibuprofen (Menac trial), as well as targeting hormonal pathways (selective androgen receptor modulators and activation of the ghrelin receptor with ghrelin agonists) [18,19,20]. Therefore, a prompt identification of sarcopenia deterioration may advise sudden utilization of aggressive support protocols.

Second, to date, no consensus exists on when to offer further TARE sessions to previously treated patients. Retreatment is usually discussed in a multidisciplinary setting and is only considered in those patients who well tolerated the first procedure without a sufficient response, not always also considering the lag of time needed for the non-invasive CT/MRI assessment window. In this clinical setting, the ability of sarcopenia status to predict patients that will suffer progression of disease may play a role in the identification of patients who might need an early TARE retreatment.

Third, the initial sarcopenic status should consent to select a sub-group of HCC patients receiving TARE as downstaging for liver transplant. Therefore, the sarcopenic status should be of great importance with the intent to “shift” these patients to a potentially curative therapy, consenting to select patients with predictable positive outcomes after transplantation [21].

Finally, in those cases not deemed to be fit for retreatment, it may anyhow be useful to promptly identify non-responder patients who may benefit from early systemic therapy [17].

The study presents some limitations. First, the study is retrospective, therefore not presenting a randomized nature. Second, the sample size analyzed is relatively small and not externally validated. However, considering the novelty of the message and the rigorous statistical approach adopted, we are confident that further investigation will be conducted on this specific setting. Further retrospective or, even better, prospective larger studies are needed. Lastly, the decision to measure sarcopenia status with the PMI measure derived from the results observed in previous studies exploring the impact of sarcopenia in the setting of HCC. Obviously, novel studies also exploring other sarcopenia measurements should be taken into account.

## 5. Conclusions

TARE treatment is nowadays a well-established therapeutic alternative for patients with locally advanced hepatocellular carcinoma. However, response to TARE treatment is usually difficult to evaluate in the early stages of follow up, thus leading to several implications, such as a delay in retreatment and systemic medical therapy. Sarcopenia status worsening at one month should represent a parameter able to identify patients at high-risk for progressive disease.

If confirmed, our results might help in an early identification of non-responder patients who may benefit from early systemic therapy, a second TARE treatment, or nutritional and physical support therapy.

However, a note of caution should be suggested in light of the small sample size of the study and more evidence is needed to support our results by larger retrospective or prospective randomized studies.

## Figures and Tables

**Figure 1 biology-10-00728-f001:**
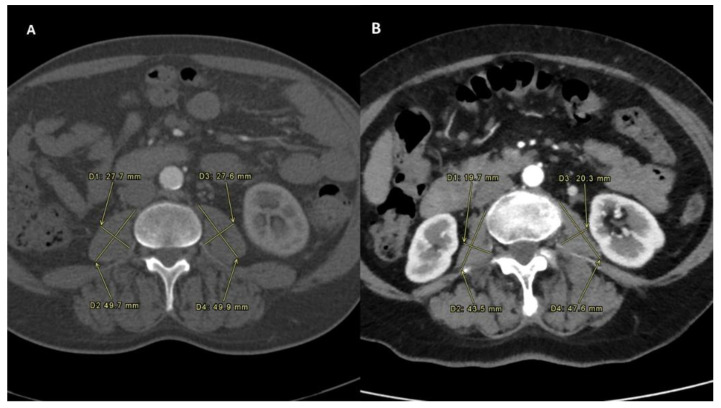
Sarcopenia status measured before the intra-arterial radioembolization in both the Sarcopenia group (**A**) and No-Sarcopenia group (**B**).

**Figure 2 biology-10-00728-f002:**
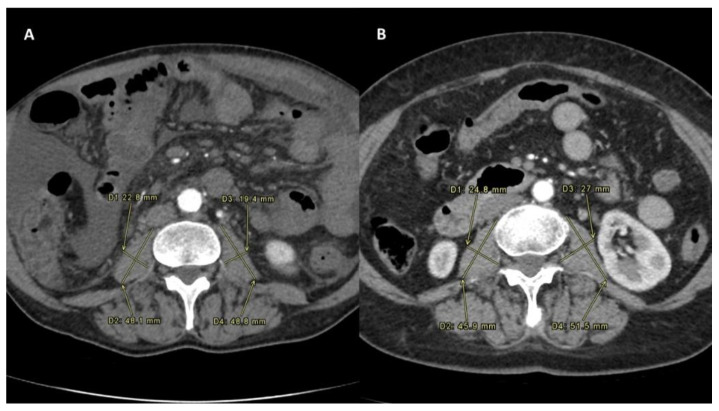
Sarcopenia status measured after the intra-arterial radioembolization in both the Sarcopenia group (**A**) and No-Sarcopenia group (**B**). All PMI measures were performed on the same level, which was between the third and the fourth lumbar vertebral body. To normalize the muscle mass for the patient’s height, this was divided by the square of the patients’ height. The PMI formula was PMI [mm/m^2^]: [(minor diameter of left psoas + major diameter of left psoas + minor diameter of right psoas + major diameter of right psoas)/4]/height in m^2^ [10].

**Figure 3 biology-10-00728-f003:**
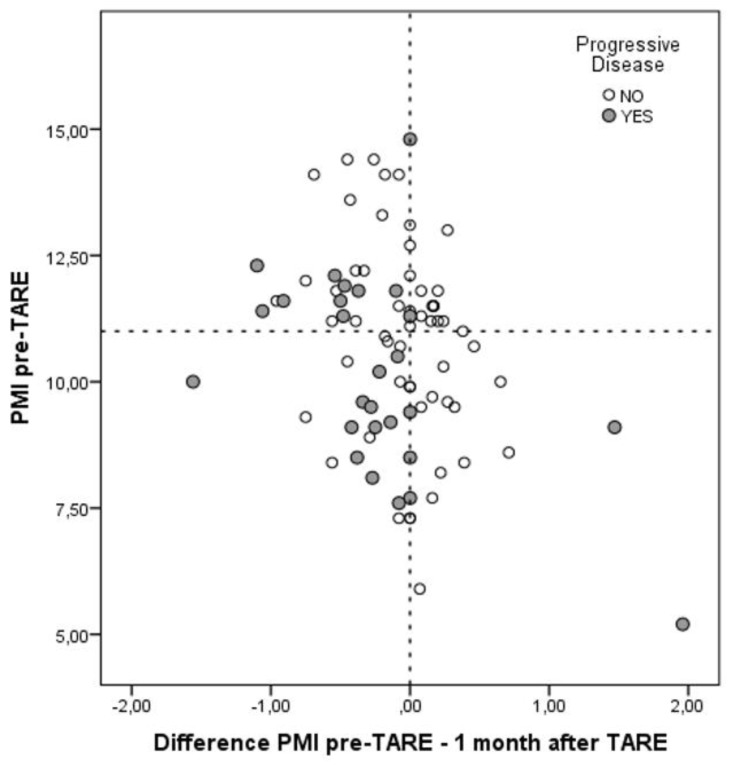
Distribution of the entire population according to the initial PMI value and the delta-PMI at one month.

**Figure 4 biology-10-00728-f004:**
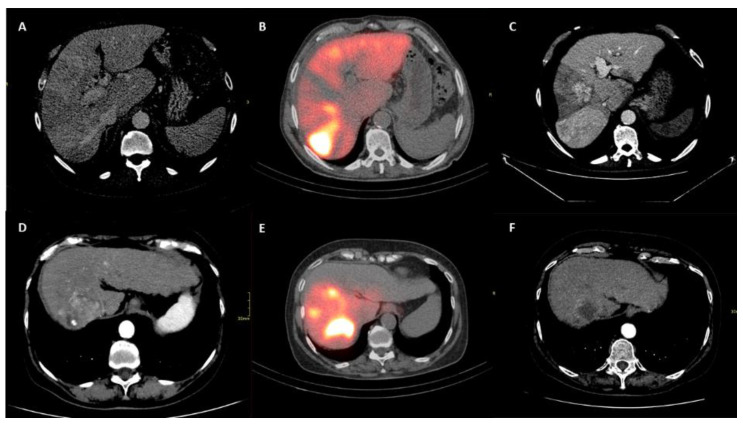
Upper row (**A**–**C**) shows patient who underwent to progression disease. Lower row (**D**–**F**) shows patient who underwent to complete response. (**A**,**D**) show liver disease on multiphasic CT before intra-arterial radioembolization. (**B**,**E**) show correctly targeted radioembolization on post-procedural SPECT–CT. (**C**,**F**) show tumor response on CT follow up.

**Figure 5 biology-10-00728-f005:**
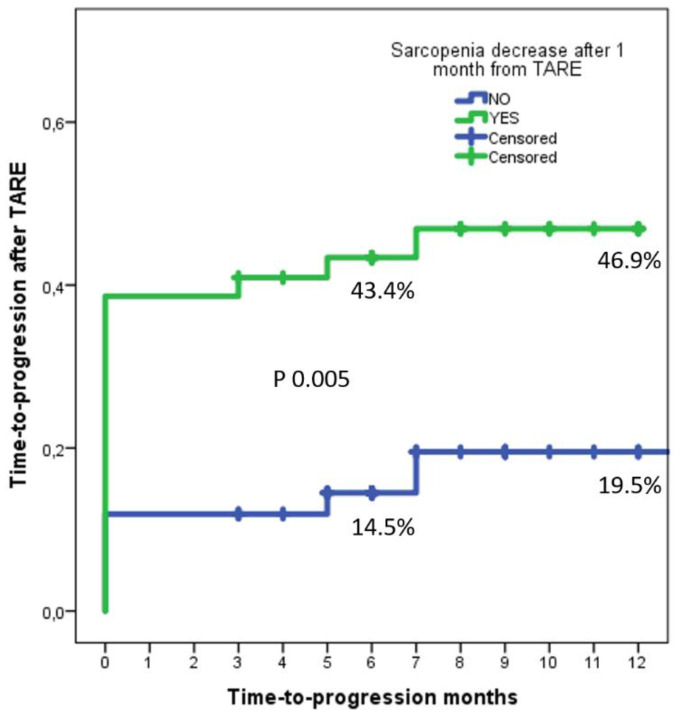
Delta-PMI at one month was statistically significant as an independent risk factor for PD, relating with a shorter time to progression.

**Table 1 biology-10-00728-t001:** Patient-, tumor- and treatment-related characteristics in the No-Sarcopenia and Sarcopenia groups.

Variable	No-Sarcopenia Group (*n* = 42)	Sarcopenia Group (*n* = 44)	*p*
	Median (IQR) or *n* (%)		
Age (year)s	64 (59–66)	64 (58–71)	0.45
Male sex	33 (78.6)	32 (72.7)	0.62
Height (cm)	176 (170–180)	173 (165–180)	0.17
Ascites (any grade) Moderate	16 (38.1) 7 (16.7)	15 (34.1) 2 (4.5)	0.82 0.09
Diameter target lesion (mm)	54 (37–77)	44 (31–79)	0.51
Number of nodules	3 (2–4)	3 (2–4)	0.48
Bilobar involvement	26 (61.9)	28 (63.6)	1.00
Liver involved >50%	12 (28.6)	9 (20.5)	0.46
Macrovascular invasion	23 (54.8)	24 (54.5)	1.00
Nodal metastases	10 (23.8)	8 (18.2)	0.60
Bilobar TARE treatment	25 (59.5)	23 (52.3)	0.52
PIVKA measure (AU/mL) at TARE	200 (93–455)	168 (44–450)	0.91
AFP measure (ng/mL) at TARE	69 (20.244)	58 (21–205)	0.87
PMI (mm/m^2^) at TARE Male sex Female sex	10.2 (8.6–11.4) 10.0 (8.6–11.6) 11.0 (8.7–11.3)	11.3 (9.5–12.1) 11.6 (10.1–12.3) 10.2 (8.7–11.5)	0.04 0.01 0.86
PMI (mm/m^2^) at 1 month Male sex Female sex	10.7 (9.2–11.5) 10.6 (9.1–11.6) 11.1 (8.8–11.4)	10.7 (9.1–11.5) 11.0 (9.9–11.9) 9.3 (8.2–10.6)	0.70 0.21 0.28
Delta PMI after 1 month Male sex Female sex	0.1 (0.0–0.3) 0.2 (0.0–0.3) 0.0 (0.0–0.2)	−0.4 (−0.5–−0.2) −0.3 (−0.5–−0.2) −0.4 (−0.8–−0.1)	<0.0001 <0.0001 <0.0001
PMI (mm/m^2^) at 3 months Male sex Female sex	10.1 (9.3–11.6) 10.0 (9.1–11.6) 11.0 (8.4–11.6)	10.4 (8.7–11.4) 10.8 (9.3–11.9) 8.9 (7.5–10.8)	0.97 0.31 0.19
Delta PMI after 3 months Male sex Female sex	0.1 (−0.3–0.4) 0.1 (−0.3–0.4) 0.0 (−0.5–0.4)	−0.6 (−1.1–−0.3) −0.6 (−1.1–−0.1) −0.8 (−1.2–−0.5)	<0.0001 0.001 0.003
Decrease PMI after 3 months Male sex Female sex	14 (33.3) 10 (30.3) 4 (44.4)	37 (84.1) 25 (78.1) 12 (100.0)	<0.0001 <0.0001 0.006
mRECIST CR	10 (23.8)	7 (15.9)	0.42
mRECIST PR	18 (42.9)	13 (29.5)	0.26
mRECIST SD	9 (21.4)	7 (15.9)	0.59
mRECIST PD	5 (11.9)	17 (38.6)	0.006
Extrahepatic PD	4 (9.5)	11 (25.0)	0.09
Intra + extrahepatic PD	2 (4.8)	8 (18.2)	0.09
PD (intra- and/or extrahepatic)	7 (16.7)	20 (45.5)	0.005

**Abbreviations:** *n*, number; IQR, interquartile ranges; TARE, transarterial radio-embolization; PIVKA, protein induced by the vitamin K absence; AFP, alpha-fetoprotein; PMI, psoas muscle index; mRECIST, modified Response Evaluation Criteria in Solid Tumors; CR, complete response; PR, partial response; SD, stable disease; PD, progressive disease.

**Table 2 biology-10-00728-t002:** Risk factors for progressive disease.

Variable	Beta	SE	Wald	HR	95%CI		*p*
					Lower	Upper	
PMI decrease 1-month after TARE	1.18	0.45	6.90	3.25	1.35	7.81	0.009
Number of nodules	0.25	0.19	1.74	1.29	0.89	1.87	0.19
Macrovascular invasion	0.50	0.46	1.16	1.65	0.66	4.08	0.28
Nodal invasion	0.46	0.47	0.97	1.59	0.64	3.96	0.32
Diameter of the target lesion	0.002	0.01	0.10	1.00	0.99	1.02	0.76

**Abbreviations:** SE, standard error; HR, hazard ratio; CI, confidence intervals; PMI, psoas muscle index; TARE, transarterial radio-embolization.

**Table 3 biology-10-00728-t003:** Diagnostic ability for progressive disease of the different sarcopenia measurements.

Variable	AUC	SE	95%CI		*p*
			Lower	Upper	
PMI at TARE	0.59	0.07	0.46	0.72	0.19
PMI after 1 month	0.65	0.06	0.53	0.77	0.03
Delta-PMI TARE—1 month	0.70	0.06	0.58	0.82	0.003
PMI after 3 months	0.73	0.06	0.63	0.84	0.001
Delta-PMI TARE—3 months	0.81	0.06	0.70	0.92	<0.0001

**Abbreviations:** AUC, area under the curve; SE, standard error; CI, confidence intervals; PMI, psoas muscle index; TARE, transarterial radio-embolization.

## Data Availability

Data used for the study are not public and aren’t available in any online database; if motivated request is received data will be provided by the authors.

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
