# Peer review of "Sarcopenia Worsening One Month after Transarterial Radioembolization Predicts Progressive Disease in Patients with Advanced Hepatocellular Carcinoma"

_biology, 2021, doi:10.3390/biology10080728_

Round 1

Reviewer 1 Report

The authors demonstrate correlation between sarcopenia modifications measured before and after one month from TARE treatment and its response rate.

It is interesting. I have some considerable concerns.

Major comment

#1. Tables and figures seem to be missing or wrongly numbered. Thus I should review this manuscript again after appropriate tables and figures are presented.

#2. The ranges of PMI in men and women have been reported to be different. The authors should present the data of male and female separately.

#3. The normal ranges of PMI are not presented.

#4. The authors used the diagnosis “sarcopenia”. But they did not present the definition of sarcopenia. I think that even sarcopenia group contains some patients without sarcopenia and that no-sarcopenia group contains some patients with sarcopenia.

#5. “Sarcopenia modification” may be changed to “skeletal muscle depletion”, since there is no sarcopenia.

Author Response

Reviewer 1

The authors demonstrate correlation between sarcopenia modifications measured before and after one month from TARE treatment and its response rate. It is interesting. I have some considerable concerns.

Response: We thank Reviewer#1 for the positive comments and for the interest in the study. 

Major comment

  1. Tables and figures seem to be missing or wrongly numbered. Thus I should review this manuscript again after appropriate tables and figures are presented.

Response: We thank Reviewer#1 for the opportunity to improve our paper. Effectively, a mistake was done in the last version of the submitted paper, completely missing to present the tables. Now, we have added the three tables, making the text coherent with the different tables and figures. 

  1. The ranges of PMI in men and women have been reported to be different. The authors should present the data of male and female separately.

Response: We agree with Reviewer#2. However, we should do some considerations on this aspect. First, the use of a “normalized” sarcopenia value deriving from the delta of two different PMI measurements completely eliminates the potential biases deriving from the different morphological dimensions between male and female patients. Second, the decision to perform inferential analyses using the entire cohort derived from the exigence to reduce the biases connected with the small sample size. Consequently, we decided to add in Table 1 the extrapolated data of PMI for men and women, and to perform a sub-analysis confirming the results in terms of time-to-progression also in the two separated sub-groups. We also did some comment in Discussion on the observed results. Interestingly, the delta PMI showed the same attitude both in men and women.  

  1. The normal ranges of PMI are not presented.

Response: The interquartile ranges of the PMI at different time-points are reported in Table 1. We did not used any PMI cut-off for defining the sarcopenic status. Instead, we used the delta value between the PMI at time of TARE and 3 months after TARE. Therefore, we did not report any cut-off obtainable from the literature.  

  1. The authors used the diagnosis “sarcopenia”. But they did not present the definition of sarcopenia. I think that even sarcopenia group contains some patients without sarcopenia and that no-sarcopenia group contains some patients with sarcopenia.

Response: We agree with Reviewer#1. Our definition of sarcopenic status was not clearly stated. We used the delta PMI values evaluated at the time of TARE and one month after TARE, defining a decrease of this value as a sarcopenic status.

Obviously, it should happen that some cases definable as “sarcopenic” using one definition should not be classified in the same way using another categorization. However, we think such a clarification in the text should surely improve the clarity of the study. 

We here report the modified sentence clarifying our definition of sarcopenia.  

“PMI was determined by measuring the major diameter and the minor diameter (measured perpendicularly to the major) of the right and of the left psoas on an axial CT scan at the time of TARE. [Figure 1] Follow-up scan after TARE therapy for the assessment of PMI modification was performed at one, three and six months after the procedure. [Figure 2] [15]

All PMI measures were performed on the same level, which was between the third and the fourth lumbar vertebral body. To normalize the muscle mass for the patient’s height it was divided by the square of the patients’ height. The PMI formula was: PMI [mm/m^2]: [(minor diameter of left psoas + major diameter of left psoas + minor diameter of right psoas + major diameter of right psoas)/4]/height in m^2. [11]

Sarcopenia status was defined as a decrease in the delta PMI calculated comparing the PMI values at the time of TARE and one month after TARE.”

  1. “Sarcopenia modification” may be changed to “skeletal muscle depletion”, since there is no sarcopenia.

Response: We modified the text according to the suggestions of Reviewer#1. 

Reviewer 2 Report

More evidence is needed to support their conclusion.

Author Response

  1. More evidence is needed to support their conclusion.

Response: We agree with Reviewer#2. We decided to modify our conclusions, adding some note of caution. Unfortunately, our results derive from a monocentric small-sized cohort. Moreover, this is the first series investigating the connection delta PMI – HCC progression after TARE. Consequently, no literature exists clearly confirming our findings. 

  We rewrote the Conclusion:

“TARE treatment is nowadays a well-established therapeutic alternative for patients with locally advanced hepatocellular carcinoma. However, response to TARE treatment is usually difficult to evaluate in early stages of follow up, thus leading to several implications such as a delay in retreatment and systemic medical therapy. Sarcopenia status worsening at one-month should represent a parameter able to identify patients at high-risk for progressive disease. 

If confirmed, our results might help in an early identification of non-responder patients who may benefit from early systemic therapy, a second TARE treatment or nutritional and physical support therapy. 

However, a note of caution should be suggested in light of the small sample size of the study, and more evidence is needed to support our results by larger retrospective or prospective randomized studies.”

Round 2

Reviewer 1 Report

The authors revised the paper according my comments. But I have some comments.

Major comments

#1. Figure 4. There is no appropriate description in the text about Fig. 4.

#2. Line 182. “Diagnostic ability of the sarcopenia measurement” should be “Diagnostic ability for progressive disease of the sarcopenia measurement” .

#3. Table 3. “Diagnostic ability of the different sarcopenia measurements” should be “Diagnostic ability for progressive disease of the different sarcopenia measurements”.

#4. Skeletal muscle depletion. They used many different words for skeletal muscle depletion such as sarcopenia decrease in fig. 5, delta-PMI in fig. 5, delta-sarcopenia at line 252, sarcopenia status difference at line 277, an early modification of sarcopenia status at line 282, and sarcopenia status deterioration at line 284. The unification may make the paper more legible.

Minor comments

#1. Line 16. “TARE” is the 1st appearance and should be spelled out.

#2. Line 18 and 21. “PMI” is the 1st appearance and should be spelled out at line 18. “PMI” at line 21 is the 2nd appearance and should not be spelled out.

#3. Line 91,“Radioembolization”. “R” should be a lower case letter.

Author Response

#1. Figure 4. There is no appropriate description in the text about Fig. 4.

We thank the reviewer and we corrected accordingly.

#2. Line 182. “Diagnostic ability of the sarcopenia measurement” should be “Diagnostic ability for progressive disease of the sarcopenia measurement” .

We thank the reviewer and we corrected accordingly.

#3. Table 3. “Diagnostic ability of the different sarcopenia measurements” should be “Diagnostic ability for progressive disease of the different sarcopenia measurements”.

We thank the reviewer and we corrected accordingly.

4. Skeletal muscle depletion. They used many different words for skeletal muscle depletion such as sarcopenia decrease in fig. 5, delta-PMI in fig. 5, delta-sarcopenia at line 252, sarcopenia status difference at line 277, an early modification of sarcopenia status at line 282, and sarcopenia status deterioration at line 284. The unification may make the paper more legible.

We thank the reviewer and we corrected accordingly.

Minor comments

#1. Line 16. “TARE” is the 1st appearance and should be spelled out.

We thank the reviewer and we corrected accordingly.

#2. Line 18 and 21. “PMI” is the 1st appearance and should be spelled out at line 18. “PMI” at line 21 is the 2nd appearance and should not be spelled out.

We thank the reviewer and we corrected accordingly.

3. Line 91,“Radioembolization”. “R” should be a lower case letter.

We thank the reviewer and we corrected accordingly.

Reviewer 2 Report

My question has been addressed.

Author Response

We thank the reviewer.